# Management of Traumatic Femur Fractures: A Focus on the Time to Intramedullary Nailing and Clinical Outcomes

**DOI:** 10.3390/diagnostics13061147

**Published:** 2023-03-17

**Authors:** Syed Imran Ghouri, Fuad Mustafa, Ahad Kanbar, Hisham Al Jogol, Adam Shunni, Ammar Almadani, Nuri Abdurraheim, Atirek Pratap Goel, Husham Abdelrahman, Elhadi Babikir, Ahmed F. Ramzee, Khalid Ahmed, Mutaz Alhardallo, Mohammad Asim, Hassan Al-Thani, Ayman El-Menyar

**Affiliations:** 1Department of Surgery, Orthopedic Surgery, Hamad Medical Corporation, Doha P.O. Box 3050, Qatar; 2Trauma Surgery Section, Hamad General Hospital (HGH), Doha P.O. Box 3050, Qatar; 3Department of Emergency, Hamad General Hospital (HGH), Doha P.O. Box 3050, Qatar; 4Clinical Research, Trauma and Vascular Surgery, Hamad Medical Corporation, Doha P.O. Box 3050, Qatar; 5Clinical Medicine, Weill Cornell Medical College, Doha P.O. Box 24144, Qatar

**Keywords:** femur fracture, shaft fracture, intramedullary nailing, bone union, orthopedic surgery and outcomes

## Abstract

Background: Femur shaft factures (FSF) are common injuries following high-energy mechanisms mainly involving motor vehicle crashes (MVC). We evaluated the timings of nailing management and analyzed the pattern of fracture union and outcome in a level 1 trauma center. Methods: This was a retrospective observational study of all the admitted trauma patients who sustained femoral fractures between January 2016 and September 2020. Data were analyzed and compared based on time to Intramedullary Nailing (IMN) (<12 h, 12–24 h and >24 h) and outcomes of FSF (union, delayed union and nonunion). Results: A total of 668 eligible patients were included in the study, of which the majority were males (90.9%) with a mean age of 34.5 ± 15.8, and 54% of the injuries were due to MVCs. The chest (35.8%) was the most commonly associated injured body region, followed by the pelvis (25.9%) and spine (25.4%). Most of femur fractures (93.3%) were unilateral, and 84.4% were closed fractures. The complete union of fractures was observed in 76.8% of cases, whereas only 4.2% and 3.3% cases had delayed union and nonunion, respectively, on the clinical follow-up. Patients in the delayed IMN (>24 h) were severely injured, had bilateral femur fracture (*p* = 0.001) and had higher rate of external fixation, blood transfusion, pulmonary complications and prolonged hospitalization. Non-union proportion was greater in those who had IMN <24 h, whereas a delayed union was greater in IMN done after 24 h (*p* = 0.5). Those with a nonunion femur fracture were more likely to have bilateral fracture (*p* = 0.003), frequently had retrograde nailing (*p* = 0.01), and high-grade femur fracture (AO type C; *p* = 0.04). Conclusion: This study showed that femur fracture is not uncommon (8.9%), which is manifested with the variety of clinical characteristics, depending on the mechanism, management and outcome in our center. Bilateral fracture, retrograde nailing and AO classification type C were the significant risk factors of non-union in patients with diaphyseal fractures. The timing of IMN has an impact on the fracture union; however, it is not a statistically significant difference. Therefore, the treating physicians should consider the potential risk factors for a better outcome by careful selection of treatment in sub-groups of patients.

## 1. Introduction

Femur factures (FF) are common injuries following high-energy mechanisms mainly involving motor vehicles crashes (MVCs) and falls from height [1]. The reported incidence rate of femoral shaft fractures (FSF) was 18.2 per 100,000 individuals per year, which often results in significant injury-related disability [2]. In polytrauma patients, concomitant injuries with FSF are the major cause of morbidity [3,4]. In such cases, FSF could be compounded by other life-threatening injuries that require urgent interventions and so the primary definitive fixation of the femur could be delayed [5]. The vast majority FSF are closed (91%), in which the surrounding tissues remain intact, whereas a lesser proportion constituted open type (9%) [1,6], presented with exposed bone, which are considered serious due to the higher risk of wound contamination and sepsis [7]. The FSF could be treated conservatively or managed surgically, by plate or intramedullary nail (IMN) fixation [2,7]. Notably, IMN is the standard of care, as it is less invasive and possesses a lower risk of in-hospital complications [8]. External fixation is indicated for complex FSF such as open fractures, vascular injuries and those with polytrauma that make early definitive care impractical [9].

Of note, the contemporary literature suggests that early surgical fixation results in a lower risk of pulmonary complications, deep vein thrombosis and death [10]. Therefore, early definitive treatment (<24 h) with antegrade reamed nailing for the fixation of FSF is the preferred treatment option that has better outcomes [11]. It reduces the risk of acute respiratory distress syndrome (ARDS), multiorgan failure and mortality [12]. However, fat and pulmonary embolism, wound infection, delayed union and non-union are considerable post-surgical complications in patients with FSF [13,14]. Although, delayed or non-union are not frequently following IMN, these significantly impacted the patient quality of life and socioeconomic wellbeing [8]. Furthermore, reaming may cause thermal necrosis and increased intramedullary pressure, which could potentially contribute to delayed healing of the fracture [15,16]. Early intensive physical therapy among young patients may result in recovery to baseline function within six months post-trauma [17,18]. On the other hand, it may take more time (12 to 24 months) to retain the recovery of strength and function among elderly population [19]. Due to the need of reoperations, and prolonged morbidity resulted in impaired functionality, nonunion remains a challenging issue to the orthopedic surgeons [20]. Therefore, it is imperative to understand the optimal timing for the stabilization of femur fracture and the outcome in young patients to improve the clinical management and rehabilitation. Herein, the present study evaluated the timing of nailing management and analyzed the outcome and pattern of fracture union in a level 1 trauma center in a rapidly developing country in the Middle East.

## 2. Materials and Methods

This retrospective observational study reviewed data for all patients with FSF admitted and treated at a level 1 trauma center, Hamad General Hospital (HGH), between January 2016 and September 2020. Inclusion criteria comprised of all adult trauma patients aged 18 years or older who had FSF and were treated at our center. Patients records were excluded if they were brought in dead, pediatric cases and unclassified as X-ray image done but films not accessible for analysis. Cases with other treatment options for proximal and distal femur fracture patterns were also excluded.

A nationally representative set of data was retrieved from the Qatar national trauma registry, which has internal and external validation to maintain high quality data. It has regular departmental quality audits and benchmarking with the American College of Surgeons Trauma Quality Improvement Program (TIQP-ACS). In our trauma center, all injured patients were assessed and treated as per the advanced trauma life support (ATLS) guidelines. The orthopedic team intervened after the clinical and radiographic confirmation of the FSF. For patients requiring IMN, the nails were mainly reamed (rIMN) and were placed in the lateral or supine position in an antegrade or retrograde technique. Ipsilateral fractures of the acetabulum, pelvis, or femoral neck and polytrauma cases that necessitate urgent simultaneous procedures mainly underwent retrograde nailing. The delayed IMN was performed in patients who had primary external fixation due to complex FSF or polytrauma requiring urgent life-saving procedures.

Data included demographic characteristics such as age, gender, nationality, mechanism of injury, pre-existing medical conditions, initial vital signs, associated injuries, abbreviated injury scores (AIS) for injured body regions, initial laboratory and radiological findings such as X-ray and Pan computerized tomographic (CT) scan, injury severity score (ISS), Glasgow coma score (GCS), blood transfusion, thromboprophylaxis, unilateral or bilateral FSF, open or closed FSF, site of fracture (proximal, diaphyseal, distal), AO classification, management, time to reamed IMN (<12 h, 12–24 h and >24 h), locking, site of entry (piriformis/trochanteric), antegrade/retrograde nailing, number of procedures, fracture outcome (union, delayed union, nonunion and lost to follow-up), hospital and ICU length of stay, in-hospital complications, and mortality.

Details of follow-up radiographic imaging performed as standard of care were retrospectively retrieved from the electronic medical records to look for the union or delayed union after 6 months and non-union after 9 months post-injury, and all images were interpreted by orthopedic surgeons. Non-union following a femoral shaft fracture was considered in the absence of radiographic evidence of union six months after the fracture [21]. An incomplete healing of a fractured femur evident clinically and radiographically after nine months is referred to as a nonunion [22]. The time to internal fixation of FSF was referred to as early if the IMN nailing was performed within 12 h of injury [23]. The Institutional Review Board (MRC-01-20-847) of the Hamad Medical Corporation granted ethical approval before starting this retrospective study with a waiver of informed consent.

Statistical analysis: When necessary, the data were shown as percentages, medians, or means with standard deviation. Comparative analyses were performed based on the time to IMN (group-1 (<12 h), group-2 (12–24 h) and group-3 (>24 h) and outcomes of FSF (union, delayed union and nonunion). The study utilized the Chi-square test to analyze differences in categorical variables and the Student’s *t*-test for continuous variables. In the case of categorical variables with expected cell frequencies below five, Yates’ corrected chi-square was applied. A statistical significance level of two-tailed *p*-value less than 0.05 was considered. The data analyses were performed using the Statistical Package for the Social Sciences version 21.0 (SPSS, Inc., Chicago, IL, USA).

## 3. Results

During the study period, there was a total of 8952 trauma admissions, of which 795 patients sustained FF (8.9%). Patient records were excluded if they were brought in dead (*n* = 43), were pediatric cases (*n* = 71) or were unclassified, as X-ray images were done but the films were not accessible for analysis (*n* = 13) (Figure 1). Therefore, the final analysis included 668 FSF patients who met the eligibility criteria. Table 1 displays the demographic and clinical characteristics of the study cohort. Most patients were males (90.9%) with a mean age of 34.5 ± 15.8, and 54% of the injuries were due to MVCs, followed by falls from height (23.8%). Diabetes mellitus (9.1%) and hypertension (8.4%) were the most common pre-existing co-morbidities. Chest (35.8%) was the most commonly associated injured body region followed by the pelvis (25.9%) and spine (25.4%). Nearly 30% had associated tibia and fibula fractures, ankle fractures were seen in an 8% and nearly 10% had concomitant knee injuries.

DVT prophylaxis was administered in 81.4% cases. Unilateral femur fractures accounted for most cases (93.3%), while closed fractures constituted 84.4% of the cases (Table 2). Seventy percent of the cases had diaphyseal fractures and had more A and B types of fractures as per AO classification. Reamed IMN was utilized in the treatment of nearly three-quarters of femur fractures, with antegrade and retrograde approaches used in 84% and 16% of cases, respectively. The pyriform fossa nails (12.0%) were seldom used, compared to trochanteric entry nails, which were used in 77% of the cases and 68 (13.8%) fractures were managed using external fixators. The fracture outcome was good with complete union in 76.8% cases, whereas 4.2% and 3.3% cases were found to have delayed union and nonunion, respectively, in the study cohort on clinical follow-up. There were 23 patients who had early mortality prior to fixation.

In half of the cases, blood transfusion was needed; the median number of blood units received was four (Table 1). After the treatment, a small percentage of 4.3% of patients experienced wound infection. The treatment for infection consisted of local wound care, excision of any devitalized tissue, nail removal, and delayed exchange of the nail when necessary. Around 9.2% of the subjects developed pulmonary complications, such pneumonia (4.5%), pulmonary embolism (2.2%), and ARDS (2.5%). The other in-hospital complications were acute renal failure due to tubular necrosis (2.1%) and sepsis (1.6%). The median length of the hospital and ICU stays were 10 and 4 days, respectively. The overall in-hospital mortality rate was 3.4%.

Table 3 demonstrates the association between the timing of IMN with outcomes, management, and complications. These groups did not differ with respect to age and gender. Patients in the delayed IMN (>24 h) group were more likely to have significantly higher ISS (*p* = 0.001), lower GCS (*p* = 0.001), bilateral femur fracture (*p* = 0.001) and had a higher rate of associated injuries mainly chest, pelvis, spine, head and abdomen (*p* = 0.001 for all). The group-2 patients had a significantly higher rate of unilateral femur fracture and had a higher frequency of type 32A (AO classification). As compared to the other groups, the rate of external fixation, blood transfusion, and in-hospital course and complications such as wound infection, ARDS and pneumonia were significantly higher in group-3 (delayed intervention). Figure 2 shows the IMN timing and union outcomes.

Table 4 compares the characteristics based on the postoperative outcomes of femoral shaft fracture. The three groups were comparable for age, co-morbidities, and associated injuries, pattern of femur fracture, time to IMN and the rate of wound infection. Those with nonunion of femur fracture were more likely to have bilateral fracture (*p* = 0.003), frequently had retrograde nailing (*p* = 0.01), and had high grade femur fracture (AO type C; *p* = 0.04) as compared to other groups. On the other hand, cases with union of fracture frequently had unilateral fractures (*p* = 0.003), had antegrade nailing (*p* = 0.005) and low-grade fractures (AO type A; *p* = 0.04). AO type B was observed in more cases who had delayed union (*p* = 0.04). Table 5 depicts the management of femur fractures treated without IMN.

## 4. Discussion

Femoral shaft fractures following trauma are often associated with short-term functional impairment, but also possesses a higher risk of long-term deformity, which compromised the daily activities of the victims. Therefore, understanding the pattern, timing of intervention and outcome of femoral fractures may assist emergency physicians and orthopedic surgeons to improve clinical management. This study outlines several key findings. It shows that the majority of femur fractures are unilateral, closed fractures and two-thirds were diaphyseal fractures treated by reamed IMN. Up on clinical follow-up, three-fourths of the patients with FSF had a favorable outcome as evidenced by bony union however, delayed union or non-union after IMN occurred in 4.2% and 3.3% cases, respectively. Moreover, delayed IMN (>24 h) was associated with severe trauma characterized by higher ISS, lower GCS, those with bilateral femur fracture and other concomitant injuries. Patients with non-union were more likely to have bilateral fractures, and had high AO classification type C, whereas AO classification type B was observed more in cases with delayed union.

There is a considerable variation in the rate of occurrence of FSF in the reviewed literature. The incidence of FSF in adults increases with advanced age owing to the age-related physiological changes [24]. Notably, diaphyseal femur fractures occur in 9.9–12 per 100,000 population per year, and around two-thirds of the affected victims are males [25]. Another study by Salminen et al. [26] reported a higher incidence of 30 per 100,000 among the younger age group (15–24 years). Similarly, the highest incidence of FSF has been reported among young individuals between the ages of 15 to 24, with a mean age of 25 years [25]. Our findings are consistent with these observations as the average age of our cohort was 35 years and the majority were males (90.9%) mainly injured secondary to high-energy MVCs (54%).

In our study, the chest and pelvis were the most frequent concomitant injuries, representing 36% and 26% of the cases, respectively. Our findings corroborate an earlier study from the Kingdom of Saudi Arabia, close to our country, which reported the head (27.5%) and chest (26%) as the most frequently injured regions among cases with FSF [27]. This study reported a 15% incidence of associated tibia shaft fractures, which corroborates with the frequency of the associated ipsilateral tibia shaft fractures (18%) in our series. However, Kuhmola et al. [28] reported an increased rate of associated chest (67%) and head (43%) injuries in the FSF patients. This could be attributed to the inclusion of severely injured patients (NISS ≥ 16) in that study.

In the present study cohort, the highest proportion of the femur fractures were closed (84.4%), mainly AO classification type 32-A, 32-B and 32-C involving the shaft of the femur in around two-thirds (70%) of the cases, which indicates high energy trauma. These findings are also in accordance with a previous study from Nigeria, showing that many patients sustained closed fractures (78%) and 56% had diaphyseal fractures [29].

To date, reamed IMN is the standard of care approach for the internal fixation of FSF primarily with the closed fractures [29,30]. An appropriate alignment, a high rate of union, lesser complications, and early mobilization are the reported benefits of stabilization with IMN [10]. In our study, nearly three-fourths of the femur fractures were treated with reamed IMN (84% were antegrade). Similarly, most of the fractures (47.2%) were fixed using locked IMN in another study [29].

Nevertheless, the appropriate time for definitive internal fixation is still a subject of debate [31].

Despite the widespread acceptance of the benefits of early femur fixation, the definition of “early definitive fixation” ranges from within 12 h to 4 days [32]. Earlier published meta-analyses have considered early definitive fixation as <24 h post-trauma and compared it with late definitive fixation after 24 h [10,33]. In the present study, patients who had delayed IMN (>24 h) were more likely to sustain severe injuries, with polytrauma and bilateral femur fracture, required more blood transfusion, had prolonged hospital course and had increased complications such as wound infection, ARDS and pneumonia. Our findings are supported by a recent meta-analysis on the timing of IMN in trauma patients [10]. The authors reported lower risk of pneumonia and ARDS in patients who had early IMN as compared to those with late IMN fixation. A prior study conducted in our institution also reported a higher rate of hospital complications and prolonged hospital course in patients with late IMN [23]. Morshed et al. [31] also found an association of worse outcomes in polytrauma patients with a delayed fixation (2–5 days) of the FSF. Recent studies have linked femur fracture treatment within 24 h of admission to better patient outcomes and shorter hospital stays [34,35]. External fixation as a means of temporary rigid stabilization in polytrauma patients with FSF is a viable and well-recognized option. It is a rapid procedure that results in minimal blood loss and can be followed by IMN once the patient’s condition becomes stable [36]. For selected polytrauma patients, a safe method for treating FSF is to perform an immediate external fixation, followed by early closed IMN [37]. However, external fixation as an initial fracture stabilization in patients with multiple injuries as a part of the damage control approach is being used less frequently [38]. External fixation is infrequently used as a treatment option for FSF, except in cases who sustained open fractures with significant soft tissue injuries [39]. In our study, an external fixator was applied in 13.8% as an initial management of the femoral shaft prior to IMN, which was subsequently converted to an IMN. Additionally, an earlier study reported a similar rate of external fixation (13.2%) in trauma patients with grade III open fracture [29].

Non-union of the FSF post definitive treatment poses a potential challenge to orthopedic surgeons. It is the failure of the femoral bone to heal due to various factors that result in pain, impaired function, delayed rehabilitation and psychological distress. The FDA defines non-union as a fracture that remains ununited for a minimum of nine months and shows no signs of healing for three consecutive months [40]. Depending on the kind of fracture and surgical approach, the non-union rate after IMN of FSF ranges between 1% and 20% [8]. In our cohort, a favorable outcome with timely radiological fracture union was achieved in 76.8% of the cases while 4.2% cases had delayed union and the non-union rate was 3.3%. Our finding is consistent with an earlier study from China that reported 2.8% femoral non-union among patients sustained closed simple fracture treated by interlock IMN [41]. Ibeanusi et al. [29] reported a lower rate of mal-union (1.0%) and non-union (1.5%) as compared to our study. The authors suggested that based on the outcomes, the adult femur fractures may be safely treated with the use of surgical techniques. Contrarily, another study reported a higher rate of nonunion (12%) among skeletally mature patients who sustained FSF and underwent IMN [42]. In our study, bilateral femur fracture, retrograde nail, and high-grade AO type C fracture were significantly associated with fracture non-union. A systematic review reported a higher association of knee pain and lesser fracture union with retrograde IMN for the management of diaphyseal fractures [43]. However, Ma et al. [41] found no such association for the rate of femur nonunion among antegrade and retrograde nails. There are certain risk factors for non-union of surgically treated traumatic diaphyseal fractures, which include open fracture, smoking, diabetes mellitus, obesity, delayed weight bearing, and fracture AO classification [44]. Delayed union in fractures has been linked to various risk factors such as AO classification type C, infraisthmal fracture, and the failure to use the reaming procedure [8]. In our study, AO type B was observed in most cases with delayed union. Estimating the amount of blood loss is crucial in making clinical decisions and providing appropriate resuscitation to prevent hemorrhagic shock and the deadly combination of coagulopathy, hypothermia, and acidosis in patients with FSF who have sustained multiple injuries. In our cohort, half of the patients required a blood transfusion and the average requirement was four units. It has been reported that patients with FSF were unlikely to require a blood transfusion during the first 48 h of hospitalization [45]. Therefore, alternative causes of bleeding should be identified in the context of polytrauma patients with hemorrhagic shock.

In the study cohort, we identified pulmonary complications such as pneumonia (4.5%), ARDS (2.5%) and pulmonary embolism (2.2%). Similarly, Kim et al. [46] also described lower frequency of pulmonary embolism (2.2%) after injury. Moreover, a meta-analysis found that performing early IMN for FSF carries a lower risk of ARDS and pneumonia, in comparison to delayed IMN fixation [10]. Therefore, in patients with multiple injuries, any reported pulmonary complications may be more likely to be associated with chest trauma rather than the IMN itself [47]. In a study involving combined femur and tibial fractures, the reported rate of surgical site infection after IMN was 11.8% [48]. In our study, however, the rate was lower at 4.3%. However, isolated femur fractures have an overall infection rate as low as 0.8% based on a study [49]. In another study that investigated high-energy FSF in young and middle-aged adults, the in-hospital mortality rate was found to be 10% [50], while in the elderly population, it was twice as high at 21% [51]. In our series, the overall mortality was 3.4%, perhaps secondary to a well-established trauma service, and coordinated interdisciplinary approach towards the management of the complex polytrauma patients. The management and outcome of FSF reflect the maturity of the trauma system [1]. To achieve successful treatment and improve the chance of joint and ambulatory function recovery, it is crucial to implement a surgical treatment promptly and reintroduce a load on the injured lower limb in a timely manner. This also decreases the risk of systemic complications that can often be fatal and helps patients to resume their daily activities with better quality of life [52].

The retrospective pattern predisposed our study to certain limitations. Although, we have drawn meaningful inferences from a large sample size, we lack information for individual patient to determine the quality of life post-injury, return to work and lost to follow-up (12.3%) as patients might be repatriated. Furthermore, selection bias may occur, because the type of operation was decided at the discretion of the treating surgeon. Third, the subgroup analyses based on the postoperative outcomes were only available for 563 patients as 23 patients had died before fixation and we lost follow-up for 82 patients and so details were not available for their outcome. Furthermore, this is a single level 1 trauma center study and so our findings may be not generalizable to other low resource trauma centers.

## 5. Conclusions

This study showed that traumatic femur fracture is not uncommon which is manifested with variety of clinical characteristics, depending on the mechanism, management and outcome in our center. Road traffic accidents continue to be a substantial contributor to femoral fractures, particularly among active young male population. Patients with delayed rIMN were more likely to be severely injured, frequently developed pulmonary complications and had a prolonged hospital course. The timing of IMN has an impact on the fracture union; however, it was not a statistically significant difference. Three-fourths of the fracture outcome were favorable with complete radiological union. Bilateral fracture, retrograde nailing and AO classification type C were the significant risk factors of non-union in patients with diaphyseal fractures. Therefore, the treating physicians should consider the potential risk factors for the worst outcomes of fracture by the careful selection of treatment in sub-groups of patients.

## Figures and Tables

**Figure 1 diagnostics-13-01147-f001:**
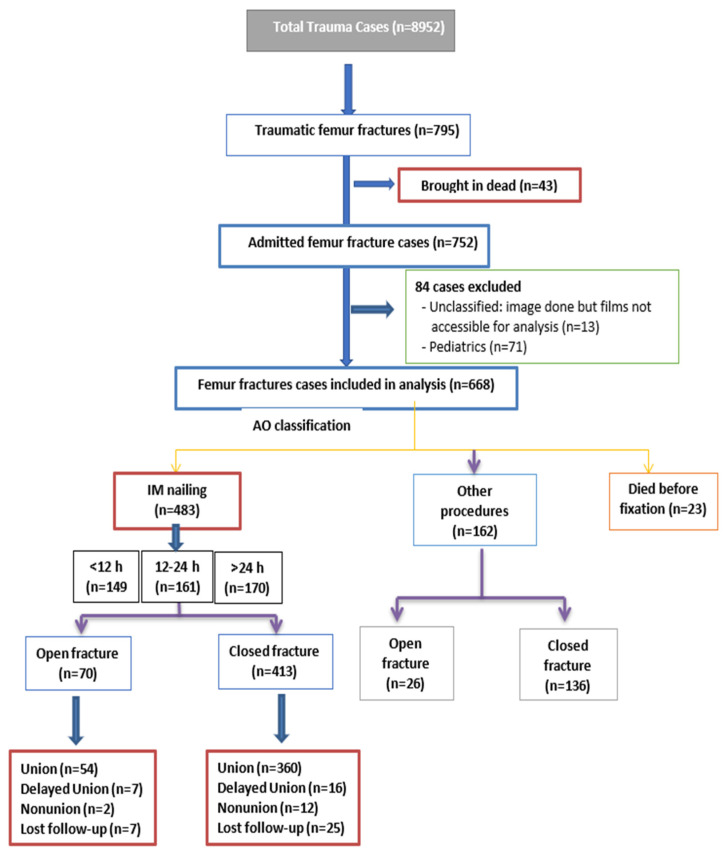
The study design.

**Figure 2 diagnostics-13-01147-f002:**
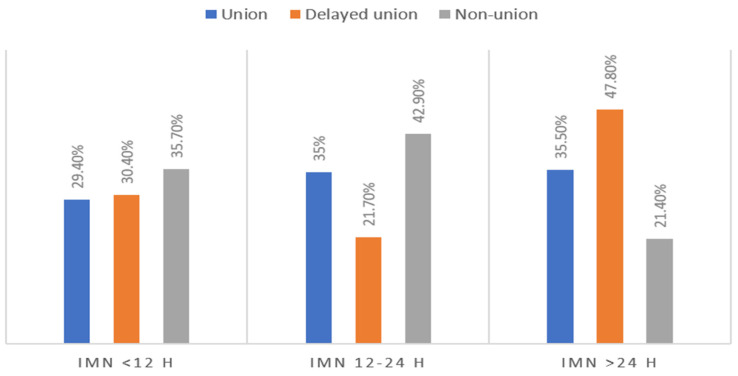
IMN timing and union outcomes.

**Table 1 diagnostics-13-01147-t001:** Demographics and clinical characteristics, complications and outcome of patients with traumatic femur fracture (January 2016–September 2020) *n* = 668.

Variables	Value	Variables	Value
Age (mean ± SD)	34.5 ± 15.8	**Abbreviated injury scores (AIS)**	
Males	607 (90.9%)	Head AIS	3.2 ± 1.0
Females	61(9.1%)	Chest AIS	2.6 ± 0.8
**Mechanism of Injury**		Abdominal AIS	2.5 ± 0.9
Road traffic accidents	359 (53.7%)	Spine AIS	2.0 ± 0.3
Fall from height	159 (23.8%)	Lower extremity AIS	3.0 ± 0.1
Pedestrian	71 (10.6%)	Pelvis AIS	2.3 ± 0.7
Fall of heavy object	41 (6.1%)	**Hemoglobin level (*n* = 509)**	12.7 ± 2.2
All-terrain vehicle	13 (1.9%)	**White blood cell count (*n* = 488)**	15.9 ± 6.9
Others	25 (3.7%)	**Neutrophil (*n* = 255)**	49.0 ± 34.6
**Co-morbidities**		**Platelets count (*n* = 499)**	247 ± 88
Diabetes mellitus	61 (9.1%)	**INR (*n* = 466)**	1.13 ± 0.16
Hypertension	56 (8.4%)	**X-ray**	621 (93.0%)
Asthma	5 (0.7%)	**Pan CT scan**	554 (82.9%)
Coronary Artery Disease	19 (2.8%)	**Blood transfusion**	334 (50.0%)
**Initial heart rate**	94.8 ± 21.4	**Blood units transfused**	4 (1–55)
**Initial systolic blood pressure**	122.4 ± 23.3	**Complications**	
**Body temperature**	36.7 ± 0.5	Pneumonia	30 (4.5%)
**Respiratory rate**	19.7 ± 5.5	Wound Infection	29 (4.3%)
**Initial Glasgow coma scale**	15 (3–15)	Acute Respiratory Distress Syndrome	17 (2.5%)
**Injury Severity Score**	16.9 ± 9.1	Pulmonary Embolism	15 (2.2%)
**Associated injuries**		Acute renal failure	14 (2.1%)
Chest	239 (35.8%)	Sepsis	11 (1.6%)
Pelvis	173 (25.9%)	**Hospital length of stay**	10 (1–185)
Spine	170 (25.4%)	**intensive care unit stay days**	4 (1–112)
Head	139 (20.8%)	**Mortality**	23 (3.4%)
Abdomen	136 (20.4%)		
Tibia	121 (18.1%)		
Fibula	71 (10.6%)		
Ankle	54 (8.1%)		
Knee	64 (9.6%)		

**Table 2 diagnostics-13-01147-t002:** Presentation and management of femur fracture cases.

Variables	Value	Variables	Value
DVT prophylaxis (*n* = 639)	520 (81.4%)	**Time to IMN; hours; (*n* = 480) ***	
Unilateral femur fracture	623 (93.3%)	<12 H	149 (31.0%)
Bilateral femur fracture	45 (6.7%)	12–24 H	161 (33.5%)
**Type of Fracture (*n* = 640)**		>24 H	170 (35.4%)
Close fracture	540 (84.4%)	**External fixation prior to IMN**	68 (13.8%)
Open fracture	100 (15.6%)	**Implant type (*n* = 435)**	
**Site of fracture**		Antegrade nail	365 (83.9%)
Proximal	231 (34.6%)	Retrograde nail	70 (16.1%)
Diaphyseal	471 (70.5%)	**Site of entry (*n* = 502)**	
Distal	100 (15.0%)	Trochanteric	385 (76.7%)
**AO classification**		Piriformis	60 (12.0%)
31A	97 (14.5%)	Retrograde	57 (11.4%)
31B	36 (5.4%)	**Locking (*n* = 489)**	457 (93.5%)
31C	16 (2.4%)	**Number of procedures (*n* = 602)**	1 (1–6)
32A	202 (30.2%)	**Fracture outcomes**	
32B	157 (23.5%)	Union	513 (76.8%)
32C	93 (13.9%)	Delayed union	28 (4.2%)
33A	23 (3.4%)	Non-union	22 (3.3%)
33B	17 (2.5%)	Lost to follow-up	82 (12.3%)
33C	27 (4.0%)	Died before fixation	23 (3.4%)
**Reamed IMN**	483 (72.3%)		

* Three patients had surgery outside the country; IMN: intramedullary nailing; DVT: deep vein thrombosis.

**Table 3 diagnostics-13-01147-t003:** Management, complications and outcome by timing of intramedullary nailing (*n* = 480).

	Time to Intramedullary Nailing			*p* Value
	Group-1 <12 h (*n* = 149)	Group-2 12–24 h (*n* = 161)	Group-3 >24 h (*n* = 170)	
**Age (mean ± SD)**	31.2 ± 11.2	31.2 ± 14.5	31.7 ± 11.9	0.78
**Males; N (%)**	137 (91.9%)	143 (88.8%)	158 (92.9%)	0.38
**Initial GCS (mean, 95%CI)**	14.1 (13.6–14.6)	14.5 (14.2–14.8)	12.5 (11.8–13.2)	0.001
**Injury Severity Score**	14.5 ± 6.8	13.8 ± 5.7	20.4 ± 9.6	0.001
**Associated injuries; N (%)**				
Chest	37 (24.8%)	41 (25.5%)	89 (52.4%)	0.001
Pelvis	26 (17.4%)	24 (14.9%)	66 (38.8%)	0.001
Spine	30 (20.1%)	31 (19.3%)	62 (36.5%)	0.001
Head	19 (12.8%)	14 (8.7%)	53 (31.2%)	0.001
Abdomen	16 (10.7%)	19 (11.8%)	62 (36.5%)	0.001
**Lower extremity AIS**	3.0 ± 0.1	3.0 ± 0.0	3.0 ± 0.0	0.10
**Pelvis AIS**	2.1 ± 0.3	2.3 ± 0.6	2.2 ± 0.5	0.46
**Head AIS**	3.05 ± 0.9	2.9 ± 0.9	3.3 ± 0.8	0.36
**DVT prophylaxis (*n* = 474)**	119 (80.4%)	132 (82.5%)	141 (84.9%)	0.56
**Unilateral femur fracture**	142 (95.3%)	158 (98.1%)	147 (86.5%)	0.001 for all
**Bilateral femur fracture**	7 (4.7%)	3 (1.9%)	23 (13.5%)	
**Type of fracture**				
Close fracture	125 (83.9%)	141 (87.6%)	144 (84.7%)	0.62 for all
Open fracture	24 (16.1%)	20 (12.4%)	26 (15.3%)	
**AO classification**				
31A	14 (9.4%)	18 (11.2%)	14 (8.2%)	0.04 for all
31B	1 (0.7%)	0 (0.0%)	2 (1.2%)	
31C	2 (1.3%)	1 (0.6%)	0 (0.0%)	
32A	60 (40.3%)	72 (44.7%)	57 (33.5%)	
32B	49 (32.9%)	48 (29.8%)	46 (27.1%)	
32C	22 (14.8%)	19 (11.8%)	41 (24.1%)	
33A	1 (0.7%)	2 (1.2%)	4 (2.4%)	
33B	0 (0.0%)	1 (0.6%)	3 (1.8%)	
33C	0 (0.0%)	0 (0.0%)	3 (1.8%)	
**Site of entry**				
Trochanteric	121 (81.2%)	125 (77.6%)	120 (70.6%)	0.23 for all
Piriformis	14 (9.4%)	20 (12.4%)	25 (14.7%)	
Retrograde	14 (9.4%)	16 (9.9%)	25 (14.7%)	
**Implant type (*n* = 429)**				
Antegrade nail	118 (85.5%)	122 (87.8%)	121 (79.6%)	0.14 for all
Retrograde nail	20 (14.5%)	17 (12.2%)	31 (20.4%)	
**External fixation prior to IMN**	9 (6.0%)	3 (1.9%)	55 (32.4%)	0.001
**Fracture outcomes**				
Union	121 (81.2%)	144 (89.4%)	146 (85.9%)	0.13 for all
Delayed union	7 (4.7%)	5 (3.1%)	11 (6.5%)	
Nonunion	5 (3.4%)	6 (3.7%)	3 (1.8%)	
**Lost to follow-up**	16 (10.7%)	6 (3.7%)	10 (5.9%)	
**Blood transfusion**	58 (38.9%)	61 (37.9%)	117 (68.8%)	0.001
**Blood units transfused**	4 (1–41)	2 (1–29)	6 (1–55)	0.001
**Complications**				
Wound infection	5 (3.4%)	0 (0.0%)	15 (8.8%)	0.001
Sepsis	1 (0.7%)	0 (0.0%)	4 (2.4%)	0.09
Pulmonary Embolism	1 (0.7%)	4 (2.5%)	8 (4.7%)	0.08
Acute Respiratory Distress Syndrome	2 (1.3%)	0 (0.0%)	8 (4.7%)	0.008
Acute renal failure	1 (0.7%)	0 (0.0%)	4 (2.4%)	0.09
Pneumonia	3 (2.0%)	3 (1.9%)	13 (7.6%)	0.009
**ICU stay; days**	4 (1–26)	2 (1–62)	7 (1–112)	0.001
**Hospital length of stay; days**	8 (1–85)	7 (2–79)	19 (1–134)	0.001

**Table 4 diagnostics-13-01147-t004:** Characteristics based on the postoperative outcomes of femoral shaft fracture.

	Union (*n* = 513)	Delayed Union * (*n* = 28)	Nonunion ** (*n* = 22)	*p* Value
**Age (mean ± SD) years**	33.6 ± 15.4	35.6 ± 13.6	36.9 ± 14.2	0.51
<50 years	445 (86.9%)	25 (89.3%)	19 (86.4%)	0.93 for all
≥50 years	67 (13.1%)	3 (10.7%)	3 (13.6%)	
**Hypertension**	42 (8.2%)	4 (14.3%)	2 (9.1%)	0.52
**Diabetes mellitus**	48 (9.4%)	2 (7.1%)	2 (9.1%)	0.92
**Injury Severity Score (ISS)**	16.5 ± 8.7	13.8 ± 5.2	14.5 ± 5.9	0.14
**ISS > 15**	206 (40.2%)	6 (21.4%)	8 (36.4%)	0.13
**Associated injuries**				
Tibia	91 (17.7%)	6 (21.4%)	4 (18.2%)	0.88
Fibula	51 (9.9%)	3 (10.7%)	2 (9.1%)	0.98
Pelvis	128 (25.0%)	6 (21.4%)	8 (36.4%)	0.43
**Unilateral femur fracture**	482 (94.0%)	24 (85.7%)	17 (77.3%)	0.003 for all
**Bilateral femur fracture**	31 (6.0%)	4 (14.3%)	5 (22.7%)	
**Type of fracture**				
Close fracture	442 (86.2%)	20 (71.4%)	18 (81.8%)	0.09 for all
Open fracture	71 (13.8%)	8 (28.6%)	4 (18.2%)	
**Implant type (*n* = 405)**				
Antegrade nail	321 (86.3%)	13 (68.4%)	9 (64.3%)	0.01 for all
Retrograde nail	51 (13.7%)	6 (31.6%)	5 (35.7%)	
**Site of entry (*n* = 466)**				
Trochanteric	333 (77.6%)	17 (73.9%)	10 (71.4%)	0.005 for all
Piriformis	56 (13.1%)	0 (0.0%)	0 (0.0%)	
Retrograde	40 (9.3%)	6 (26.1%)	4 (28.6%)	
**Reamed IMN**	414 (80.7%)	23 (82.1%)	14 (63.6%)	0.14
**Site of fracture**				
Proximal	172 (33.5%)	9 (32.1%)	8 (36.4%)	0.95
Diaphyseal	383 (74.7%)	21 (75.0%)	14 (63.6%)	0.50
Distal	60 (11.6%)	6 (21.4%)	5 (22.7%)	0.11
**AO classification**				
31A	69 (13.5%)	2 (7.1%)	4 (18.2%)	0.04 for all
31B	24 (4.7%)	3 (10.7%)	2 (9.1%)	
31C	13 (2.5%)	0 (0.0%)	0 (0.0%)	
32A	174 (33.9%)	4 (14.3%)	2 (9.1%)	
32B	122 (23.8%)	11 (39.3%)	7 (31.8%)	
32C	73 (14.2%)	4 (14.3%)	4 (18.2%)	
33A	13 (2.5%)	3 (10.7%)	0 (0.0%)	
33B	12 (2.3%)	0 (0.0%)	1 (4.5%)	
33C	13 (2.5%)	1 (3.6%)	2 (9.1%)	
**Time to IMN (hours)**				
<12	121 (29.4%)	7 (30.4%)	5 (35.7%)	0.50 for all
12–24	144 (35.0%)	5 (21.7%)	6 (42.9%)	
>24	146 (35.5%)	11 (47.8%)	3 (21.4%)	
**External fixation prior to IMN**	61 (14.6%)	4 (17.4%)	1 (7.1%)	0.68
**Wound Infection**	22 (4.3%)	2 (7.1%)	2 (9.1%)	0.46
**Hospital length of stay**	10 (1–185)	8 (2–127)	10.5 (2–129)	0.32
**ICU LOS**	4 (1–112)	3 (1–16)	2 (1–30)	0.11

* Follow-up for 6 months; ** followed-up for 9 months.

**Table 5 diagnostics-13-01147-t005:** Management approach for femur fracture (non-IM nailing cases) *n* = 162.

Open Fractures (*n* = 26)	Number of Procedures
Open reduction and internal fixation (ORIF)	7
External fixation	5
Plate and screw fixation	6
External fixation, screw and plate	2
External fixation and wound debridement	1
Plate cementing	1
Wound debridement and exploration	1
ORIF and wound debridement	1
Skeletal traction	1
Vascular repair	1
**Closed fractures (*n* = 136)**	
Conservative	31
Open reduction and internal fixation	20
Plate and Screw	18
Closed reduction	16
Screw fixation	13
Dynamic Hip Screw (DHS)	12
External fixation	7
Hemiarthroplasty and DHS	5
Skeletal traction	4
Plate and wire	2
External fixation and ORIF	2
Screw fixation	1
K-wire and screw fixation	1
Peri-loc plating	1
Cannulated hip screw fixation of femur neck	1
Cemented left hip Bipolar hemiarthroplasty	1
Calcaneal fracture fixation	1

## Data Availability

All data generated or analyzed during this study are included in this article. Data are accessible upon agreement with the Medical Research Centre, Qatar.

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
