# Peer review of "Management of Traumatic Femur Fractures: A Focus on the Time to Intramedullary Nailing and Clinical Outcomes"

_diagnostics, 2023, doi:10.3390/diagnostics13061147_

Round 1
Reviewer 1 Report
Dear Authors, I consider your retrospective observational study exposing how IMN with reaming is the standard approach for internal fixation of the FSF, mainly for closed fractures, very interesting. This is in the face of clear advantages achievable through proper alignment of fracture stumps, low risk of complications, high rate of skeletal healing and early functional recovery. On this point, let me point out for your information a personal contribution (DOI 10.7417/CT.2022 . 2453), where it is emphasized that femur fractures represent one of the main causes of hospitalization of elderly patients, which is not always easy to manage perioperatively, which is why appropriate monitoring is recommended in order to prevent and - where possible - avoid the occurrence of predictable complications (infections, pressure sores, bronchopneumonia, mental and general condition decline), reducing the mortality rate, with the intention of facilitating functional recovery, the resumption of ambulation, the return to daily life and in particular the reintegration into the family environment. In fact, as now acquired in the historical concepts of traditional orthopedics, the shortest possible delay in surgical treatment and the quickest possible resumption of load on the injured lower limb, represent prerequisites of great importance for therapeutic success, with containment of dreaded systemic complications, often fatal, and greater chances of recovery of joint and ambulation function, essential prerequisites to facilitate the resumption of the common activities of daily life. I would consider it appropriate to request you to contemplate this literature reference, which is entirely in line with your technical findings.
I also appreciated the review of the literature, which seems to to be very detailed, although limitations due to the retrospective design remain evident, as we do not have information on individual patients to determine quality of life after injury, return to work and losses at follow-up. However, I would like to congratulate for your exposition; the study certainly has interesting points as it deals with a topic of great interest. The purpose is clear and respected. The information provided is sufficient and presents useful elements to encourage the development of new scientific work. The paper is well written, following established scientific assumptions. The study design is rational. The conclusions presented are meaningful and the statistical sample used is correct.
Author Response
Thanks for your comments that fortify our paper. The suggested reference has been added.
Reviewer 2 Report
The paper presents a retrospective observational study for patients who were admitted with femoral fractures focusing on the Time to Intramedullary Nailing and Clinical Outcomes.
The study seems of not general interest and does not contain any contribution that, to the reviewer opinion, can be classified as an original contribution of scientific interest.
Overall, taking into account the topic of the special issue, the paper can be accepted for publication in its present form.
Author Response
Thanks for your comment. We do believe that sharing our experience with the readers and experts will be useful in the field of trauma particularly with a relatively good sample size and detailed analyses.